# STEED: A data mining tool for automated extraction of experimental parameters and risk of bias items from *in vivo* publications

**Wolfgang Emanuel Zurrer**[1‡], **Amelia Elaine Cannon**[1‡], **Ewoud Ewing**[2], **David Brüschweiler**[1], **Julia Bugajska**[1], **Bernard Friedrich Hild**[1], **Marianna Rosso**[1], **Daniel Salo Reich**[3], **Benjamin Victor Ineichen**[1,2,3]*

**1** Center for Reproducible Science, University of Zurich, Zurich, Switzerland, **2** Department of Clinical Neuroscience, Center for Molecular Medicine, Karolinska University Hospital, Karolinska Institute, Stockholm, Sweden, **3** Translational Neuroradiology Section, National Institute of Neurological Disorders and Stroke, National Institutes of Health, Bethesda, MD, United States of America

‡ WEZ and AEC are contributed equally to this work as share first authorship.
* benjamin.ineichen@uzh.ch

**Data Availability Statement:** The datasets generated and/or analysed during the current study are available on the Open Science Framework (OSF): https://osf.io/n8dz7/. All included

## Abstract

### Background and methods

Systematic reviews, i.e., research summaries that address focused questions in a structured and reproducible manner, are a cornerstone of evidence-based medicine and research. However, certain steps in systematic reviews, such as data extraction, are labour-intensive, which hampers their feasibility, especially with the rapidly expanding body of biomedical literature. To bridge this gap, we aimed to develop a data mining tool in the R programming environment to automate data extraction from neuroscience *in vivo* publications. The function was trained on a literature corpus (n = 45 publications) of animal motor neuron disease studies and tested in two validation corpora (motor neuron diseases, n = 31 publications; multiple sclerosis, n = 244 publications).

### Results

Our data mining tool, STEED (STructured Extraction of Experimental Data), successfully extracted key experimental parameters such as animal models and species, as well as risk of bias items like randomization or blinding, from *in vivo* studies. Sensitivity and specificity were over 85% and 80%, respectively, for most items in both validation corpora. Accuracy and F1-score were above 90% and 0.9 for most items in the validation corpora, respectively. Time savings were above 99%.

### Conclusions

Our text mining tool, STEED, can extract key experimental parameters and risk of bias items from the neuroscience *in vivo* literature. This enables the tool's deployment for probing a field in a research improvement context or replacing one human reader during data

publications to develop and validate the data mining tool are reported in the supplementary reference list.

**Funding:** This work was supported by grants of the Swiss National Science Foundation (No. 407940_206504, to BVI), the UZH Alumni (to BVI), and the Intramural Research Program of NINDS. We thank all our funders for their support. The funders had no role in study design, data collection and analysis, decision to publish, or preparation of the manuscript.

**Competing interests:** The authors report no competing interests related to this study.

extraction, resulting in substantial time savings and contributing towards the automation of systematic reviews.

## Introduction

Synthesising evidence is an essential part of scientific progress [1]. To this end, systematic reviews—i.e. the rigorous identification, appraisal, and integration of all available evidence on a specific research question—have become a default tool in clinical research [2, 3]. Yet, they are also increasingly employed for preclinical *in vivo* research [4–7].

Systematic reviews allow the identification of trends that may be missed when reviewing individual, smaller studies, and add soundness to one's conclusions. For this reason, the use of systematic reviews in animal research is an acknowledged aid to implementing the reduction, replacement, and refinement of animal experiments [8], e.g., by gaining knowledge without the use of new animal experiments or by improving the ethical position of animal research by increasing the value and reliability of research findings [9].

The process of manual evidence synthesis is highly laborious [10]. This problem is further hampered by the skyrocketing amount of publications in the biomedical field [11] and these numbers are set to increase still further in the near future [12]. With this, it becomes increasingly difficult to keep abreast with the published evidence which in turn precludes evidence-based research [13]. Consequently, automation of the labour-intensive steps of a systematic review is warranted to optimize the value of published data in the age of information overload. One particularly labour-intensive systematic review task which would profit from automation is data extraction [14, 15], i.e., the manual retrieval of specific data from publications. Based on these shortcomings, we set out to develop a text mining tool to automatically extract key study parameters from publications of animal research modelling motor neuron diseases and multiple sclerosis. Our endeavour is focused on two key domains of experimental science, that is 1) disease model parameters such as animal models and species, and 2) risk of bias measures such as randomization or blinding.

## Methods

### Study protocol

The development of the text mining tool was part of a systematic review on neuroimaging findings in motor neuron disease animal models registered as prospective study protocol in the International Prospective Register of Systematic Reviews (PROSPERO, CRD42022373146).

### Literature corpora

Three literature corpora were included in this study: one for the training of the text mining toolbox and two for its validation. The training corpus was identified by searching Medline via PubMed for animal motor neuron disease models using the search string: *"motor neuron disease" OR motor neuron diseases [MeSH] OR "amyotrophic lateral sclerosis" OR "ALS" OR "MND" OR "SOD"* and limiting the search to the publication year 2021. The two validation corpora are derived from two in-house systematic reviews: a systematic review on neuroimaging findings in motor neuron disease animal models [16] and a systematic review on neuroimaging findings in multiple sclerosis animal models [17].

## Parameters to extract and development of text mining tool

We defined items of interest to extract a priori which belong to two domains: first, experimental parameters including 1) animal species, 2) animal sex, 3) model disease, 4) number of experimental animals used, and 5–7) experimental outcomes, i.e., whether a respective study assessed behavioral, histological, or neuroimaging outcomes. Second, risk of bias items including: 1) implementation in the experimental setup of any measure of randomization, 2) any measure of blinding, 3) prior sample size calculation (power calculation), 4) statement of whether conducted animal experiments are in accordance with local animal welfare guidelines, 5) statement of a potential conflict of interest, and 6) accordance with the ARRIVE guidelines [18]. This second domain also includes an item for the data availability statement, i.e., a statement whether and where primary study data are available.

For each item of interest, we developed a library of regular expressions (RegEx) in the R programming environment. RegEx are patterns of characters that define specific text matches. This library was built by methodically gathering relevant words and phrases from the training corpus. Notably, only one study in our training corpus reported neuroimaging outcomes, prompting us to enrich our RegEx library with terms from another unpublished animal systematic review. We aimed to minimize overfitting by avoiding hard-coded expressions, yet some unique terms were essential to include to the RegEx libraries.

Using the RegEx libraries, we created an R function to extract data from scientific papers. This process starts with converting PDFs to text using the 'pdftools' package and then applying the 'stringr' package to identify relevant RegEx patterns. The function segments each paper into sections (like results or methods), strips the 'references' section, searches for matching RegEx patterns, and then aggregates this data into a dataframe. Each paper corresponds to one row in the dataframe, with columns representing the different data points extracted.

The RegEx libraries and the R function were iteratively improved to maximize performance, based on a pre-defined threshold (see below). Both our RegEx libraries and the R function are available at: https://github.com/Ineichen-Group/Auto-STEED or on the Open Science Framework (OSF): https://osf.io/n8dz7/.

## Assessment of text mining tool performance

Performance of our text mining function was gauged using the following metrics:

$$Sensitivity \ = \ \frac{TP}{TP + FN} \ (1)$$

$$Specificity \ = \ \frac{TN}{TN + FP} \ (2)$$

$$Precision \ = \ \frac{TP}{TP + FP} \ (3)$$

$$Accuracy \ = \ \frac{TP + TN}{TP + TN + FP + FN} \ (4)$$

$$F1 - score \ = \ \frac{2*TP}{2*TP + FP + FN} \ (5)$$

With TP, TN, FP, and FN being true positive, true negative, false positive, and false negative, respectively. We used R to calculate these performance metrics.

All included literature corpora have undergone dual and independent manual extraction of these parameters (WEZ, AEC, BVI) constituting the 'gold standard' for data extraction. We measured mean extraction time for both the human and the automated extraction to gauge time savings by the automated extraction. As defined in the protocol, for development of the text mining function in the training set, automated extraction of individual items was considered to be sufficiently accurate if they attained a sensitivity of 85% and a specificity of 80% (i.e., with a slightly higher sensitivity as per recommendation by the 'Systematic Living Information Machine' [SLIM] consortium).

## Results

### General characteristics of literature corpora

We included three literature corpora with manual annotation by two trained and independent reviewers. The training corpus comprised 45 individual publications on motor neuron disease animal models from 2021. The validation sets included 31 publications on neuroimaging in motor neuron disease animal models and 244 publications on neuroimaging in multiple sclerosis animal models, with median publication years of 2014 and 2009, respectively (see S1 File).

The median reporting prevalence for experimental parameters was 85%, 95%, and 93% in the training and validation corpora, respectively. Similarly, the median reporting prevalence for risk of bias items was 58%, 19%, and 20% in the training and validation corpora, respectively. A detailed summary of the characteristics and reporting prevalence of the literature corpora is presented in Table 1.

The interrater agreement was 85–95% for experimental parameters and 81–100% for risk of bias items in the training and validation corpora.

### Architecture of text mining tool

Due to copyright restrictions on data mining from HTML, the tool was developed for extracting data at the PDF publication level. Initially, the text mining function reads in PDFs of the relevant publications and converts them to text. This text is then cleaned of certain keywords, such as 'random primer,' to reduce false positives for items we aim to extract, like randomization. Subsequently, the manuscript's body is parsed into different sections (e.g., abstract, introduction, materials, and methods) based on the appearance of specific RegEx, such as the heading 'materials and methods.' Then, specific sections of the paper are mined for relevant regular expressions, using RegEx libraries tailored to each item that needs to be extracted. More concretely, the function extracts experimental parameters as well as some risk of bias items (randomization, blinding, and animal welfare statement) from the methods section and the other risk of bias items from the entire manuscript (excluding the 'references' section). The mining pipeline is depicted in Fig 1. The tool can be accessed directly on Github at https://github.com/Ineichen-Group/Auto-STEED.

PDFs of full texts are imported into the R environment, converted to text, and cleaned. Subsequently, the text is parsed into different sections such as 'materials and methods' or 'results'. Then, individual items to mine are extracted using custom-made Regex libraries and a data frame with the extracted items is created.

### Performance metrics of STEED

In the training set, the text mining function was tuned until it reached a sensitivity of 85% and a specificity of 80% for each individual item. The specificity threshold was not attained for the

**Table 1. Characteristics of included literature corpora and reporting prevalence for parameters to extract.**

| | Training corpus | Validation corpus 1 | Validation corpus 2 |
|---|---|---|---|
| **Characteristics of eligible publications** | | | |
| Topic | Motor neuron disease animal models | Neuroimaging in motor neuron disease animal models | Neuroimaging in multiple sclerosis animal models |
| Number of publications | 45 | 31 | 244 |
| Publication year median and range | 2021 (2021–2021) | 2014 (2004–2020) | 2009 (1985–2017) |
| Number of different journals | 35 | 22 | 72 |
| **Reporting prevalence** | | | |
| Experimental parameters: | | | |
| Species | 100% | 100% | 100% |
| Sex | 87% | 61% | 78% |
| Model | 100% | 100% | >99% |
| Outcome histology | 82% | 90% | 85% |
| Outcome behaviour | 73% | 42% | 61% |
| Outcome imaging | 2% | 100% | 100% |
| Risk of bias items: | | | |
| Randomization | 58% | 23% | 20% |
| Blinding | 47% | 19% | 32% |
| Animal welfare | 98% | 90% | 78% |
| Conflict of interest | 96% | 58% | 25% |
| Sample size calculation | 27% | 10% | 1% |
| ARRIVE guidelines | 29% | 0% | 1% |
| Data availability | 69% | 19% | 2% |

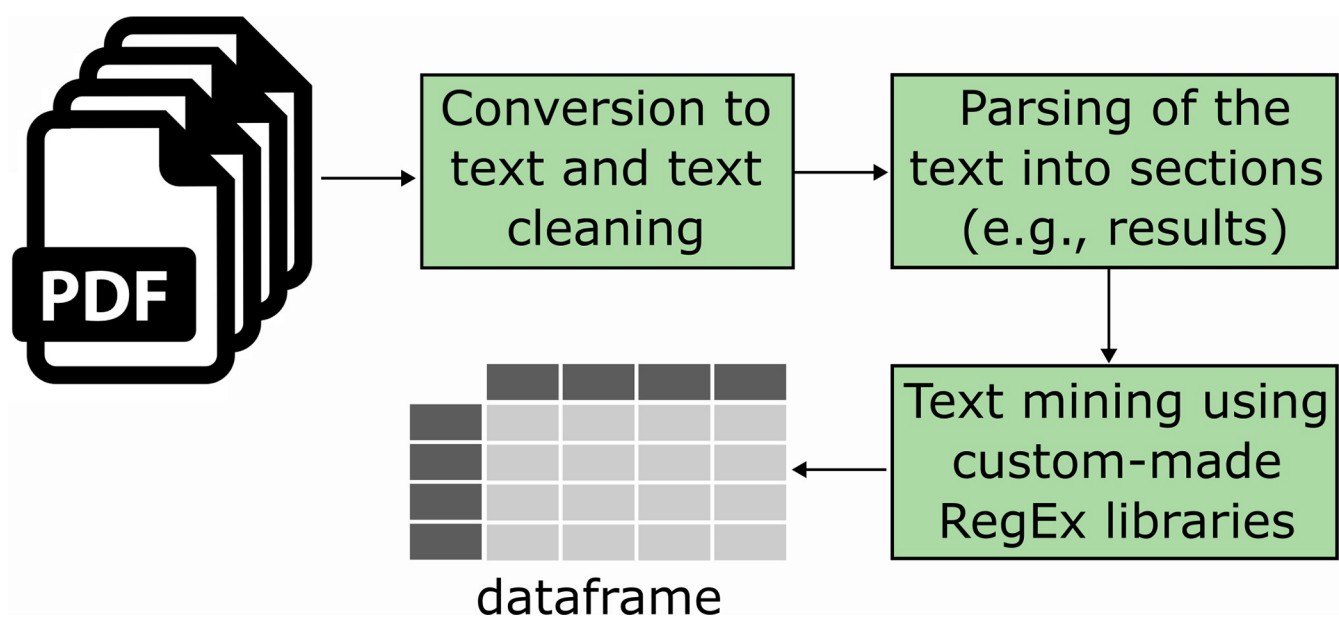

**Fig 1. Architecture of the text mining function.**

items 'sample size calculation', 'sex', and 'outcome behaviour' with only 78%, 67% and 50%, respectively but with above-threshold sensitivity. Some items such as accordance with the ARRIVE guidelines or whether a conflict-of-interest statement was included reached a sensitivity close to 100%. F1-score and accuracy were above 90% for most items (**Table 2**).

The mining function performed well on both validation corpora. In the motor neuron disease corpus, the mining function accomplished above-threshold specificity and sensitivity for most items, except for 'outcome behaviour' with slightly below-threshold specificity and 'data availability', 'sample size calculation', and 'sex' with slightly below-threshold sensitivity. In the multiple sclerosis validation corpus, additional items did not reach the specificity and sensitivity thresholds. However, F1-score and accuracy were above 90% for most items in the motor neuron disease validation corpus and above 80% in the multiple sclerosis corpus, respectively (**Table 2**).

### Time savings automated versus manual extraction

Mean time for the manual extraction was 12 (± standard deviation: 8), 13 (± 7), and 15 (± 11) minutes per publication and per human reader for the training corpus and the two validation corpora, respectively. This amounts to a total of 540, 403, and 3660 minutes for one reader for the three corpora, respectively. In contrast, the mining function required 0.3 seconds to mine one record amounting to 0.23, 0.15, and 1.22 minutes for the three corpora. With this, the text mining function provides time savings above 99%.

### Reporting of items on abstract versus full text level

For the experimental parameters, we quantified how commonly the respective items were reported in the abstract in addition to the full text. Disease models and species as well as outcome measures were commonly reported on abstract level in all three literature corpora with reporting frequencies between 95–100%. However, animal sexes were only rarely reported with reporting frequencies between 0 and 5%.

## Discussion

### Main findings

We developed STEED (STructured Extraction of Experimental Data), an R-based text mining tool designed to automatically extract key experimental details, such as animal models and species, and risk of bias factors like randomization or blinding, from preclinical *in vivo* studies. The tool demonstrated high sensitivity, specificity, and accuracy for extracting most items across two validation literature corpora. These corpora included one in a field similar to the training set (motor neuron diseases) and another in a different area (multiple sclerosis), both encompassing older publications as well. The use of STEED substantially reduced the time required to extract these data.

### Findings in the context of existing evidence

STEED performed well on literature corpora outside of the field is has been developed in as well as in corpora with older publication years, i.e., it has been developed in a corpus covering the motor neuron disease literature and performed well in a corpus of the multiple sclerosis literature. Thus, our developed function could be applied to literature bodies of other research fields. However, adapting STEED to new disciplines requires some consideration: While the tool has shown flexibility across related fields, creating discipline-specific versions may necessitate refining the underlying RegExes to accurately capture more distinct experimental

**Table 2. Summary of performance measures of STEED compared with manual human ascertainment.**

| | Specificity | Sensitivity | Precision | Accuracy | F1-score |
|---|---|---|---|---|---|
| **Training corpus (motor neuron diseases, n = 45)** | | | | | |
| Species | *NA* | **96** | 100 | 96 | 0.98 |
| Sex | 67 | **85** | 94 | 82 | 0.89 |
| Disease model | *NA* | **96** | 100 | 96 | 0.98 |
| Outcome histology | **89** | **92** | 97 | 91 | 0.94 |
| Outcome behaviour | 50 | **97** | 84 | 84 | 0.90 |
| Outcome imaging | **96** | *NA* | *NA* | 96 | *NA* |
| Randomization | **84** | **96** | 89 | 91 | 0.93 |
| Blinding | **95** | **92** | 96 | 93 | 0.94 |
| Animal welfare | *NA* | **86** | 97 | 84 | 0.92 |
| Conflict of interest | **100** | **98** | 100 | 97 | 0.99 |
| Sample size calculation | 78 | **92** | 63 | 82 | 0.75 |
| ARRIVE guidelines | **100** | **100** | 100 | 100 | 1.00 |
| Data availability | **85** | **94** | 94 | 91 | 0.94 |
| **Validation corpus 1 (motor neuron diseases, n = 31)** | | | | | |
| Species | *NA* | **100** | 100 | 100 | 1.00 |
| Sex | **100** | 74 | 100 | 84 | 0.85 |
| Disease model | *NA* | **90** | 100 | 90 | 0.95 |
| Outcome histology | **100** | **96** | 100 | 97 | 0.98 |
| Outcome behaviour | 78 | **85** | 76 | 81 | 0.79 |
| Outcome imaging | NA | **100** | 100 | 100 | 1.00 |
| Randomization | **100** | **86** | 100 | 97 | 0.92 |
| Blinding | **100** | **89** | 100 | 97 | 0.94 |
| Animal welfare | **100** | **89** | 100 | 90 | 0.94 |
| Conflict of interest | **92** | **94** | 94 | 94 | 0.94 |
| Sample size calculation | **81** | 80 | 44 | 81 | 0.57 |
| ARRIVE guidelines | **100** | *NA* | *NA* | 100 | *NA* |
| Data availability | **96** | 83 | 83 | 94 | 0.83 |
| **Validation corpus 2 (multiple sclerosis, n = 244)** | | | | | |
| Species | *NA* | 75 | 100 | 75 | 0.86 |
| Sex | 76 | 83 | 93 | 82 | 0.88 |
| Disease model | *NA* | **87** | 100 | 88 | 0.93 |
| Outcome histology | 64 | **96** | 93 | 91 | 0.95 |
| Outcome behaviour | 66 | **91** | 81 | 82 | 0.86 |
| Outcome imaging | *NA* | **94** | 100 | 94 | 0.97 |
| Randomization | **93** | 81 | 75 | 90 | 0.78 |
| Blinding | **98** | 85 | 96 | 93 | 0.90 |
| Animal welfare | **86** | 80 | 95 | 82 | 0.87 |
| Conflict of interest | **96** | **97** | 90 | 97 | 0.93 |
| Sample size calculation | **94** | **100** | 27 | 97 | 0.43 |
| ARRIVE guidelines | **100** | **100** | 100 | 100 | 1.00 |
| Data availability | **100** | 80 | 80 | 100 | 0.80 |

Specificity, sensitivity, precision, and accuracy are denoted in percentage. For details regarding measures, please see the materials and methods section. Items reaching or exceeding our pre-defined thresholds (sensitivity of 85% and a specificity of 80%) are printed in bold font.

parameters pertinent to each field. This process would involve collaborative efforts with domain experts to ensure the tool's precision and subsequent validation [19]. Consequently, while separate packages for each discipline are conceivable, they would require some adaptation efforts to maintain STEED's standards of accuracy and utility.

Although STEED showed relatively high performance, it is not yet ready for evaluating individual publications and cannot completely replace manual data extraction. Nevertheless, this automated tool has two practical applications: first, it can be employed to large reference libraries (over 1000 records) to survey specific fields for experimental parameters and potential biases [20]. Secondly, STEED can serve to replace one human reviewer during the data extraction of e.g., a systematic review, which would still lead to substantial labour savings [15, 21]. Any discrepancies between human and machine analysis can be manually reviewed for accuracy.

Similar approaches have been leveraged to extract specific information—such as the study population, intervention, outcome measured and risks of bias—from abstracts [22] or full texts [20, 23]. Bahor and colleagues developed a text mining function in a literature body of stroke animal models able to extract certain risk of bias items including randomization, blinding, and sample size calculation [24]. The achieved accuracy was between 67–86% for randomization (our approach: 90–97%), 91–94% for blinding (our approach: 93–97%), and 96–100% for sample size calculation (our approach: 81–97%). With this, our developed tool shows similar performance metrics and does complement former tool by extracting additional risk of bias items such as statement of a conflict of interest, accordance with local animal welfare regulations, a data availability statement, and accordance with the ARRIVE guidelines [18]. Another text mining toolbox based on natural language processing (NLP) was developed by Zeiss and colleagues [22]: This toolbox extracts data such as species, model, genes, or outcomes from PubMed abstracts with F1-score between 0.75 and 0.95.

For many tasks, NLP models seem to outperform RegEx-based text mining [11, 25]. Yet they are more complex and labour-intensive to develop and deploy and thus only warrant application in more complex extraction tasks. Wang and colleagues tested performance of a variety of models such as convolutional neural networks to extract risk of bias items from preclinical studies [20]. These models outperformed RegEx-based methods for four risk of bias items with F1-score between 0.47–0.91. The validity of NLP for such tasks has also been corroborated by SciScore—a proprietary NLP tool that can automatically evaluate the compliance of publications with six rigour items taken from the MDAR framework and other guidelines [23]. These items mostly relate to risk of bias, including compliance with animal welfare regulations, blinding/randomisation, prior sample size calculation and other items such as organism or animal sex. SciScore was developed on a training corpus from PubMed open access articles. In contrast, our approach was developed on preclinical neuroscience corpora thus being more tailored to this field. Additionally, techniques involving generative large language models like GPT have been explored to automate data extraction from systematic reviews [26]. While these methods show promise, they require further evaluation to establish reliability. Current findings indicate that such models may extract incorrect data [27]. Furthermore, these models often face challenges in extracting key information and tend to be more prone to errors, especially when summarizing extensive text.

While our original plan included extracting the number of animals used in studies, we had to abandon this objective due to the highly heterogeneous ways these numbers are reported— such as in the methods/results sections, tables, figure legends, graphs, or only separate for experimental and control groups. A possible approach to address this issue could be to treat it as an NLP categorization task, classifying studies into small (for instance, fewer than 10 animals), medium (10–50 animals), and large groups (more than 100 animals).

## Limitations

Firstly, our method was developed and tested specifically for preclinical neuroscience research. Its effectiveness in other areas, such as *in vivo* cancer studies, is yet to be determined. Secondly, our tool relies on full-text PDFs for data extraction. While extracting data from online versions of publications (HTML format) could solve problems related to PDF conversion, such as inconsistent layouts and varying journal formats, current copyright regulations and the need for costly licenses make this challenging [28]. Lastly, while our automated approach offers substantial time savings compared to manual data extraction, this does not take into account the time needed to verify the results of the automated process.

## Conclusions

Our developed text mining tool STEED is able to extract key risk of bias items and experimental parameters from the neuroscience *in vivo* literature. Accelerating the usually labour-intensive data extraction during a systematic review contributes towards automation of systematic reviews.

## Supporting information

**S1 File. Supplementary reference list.**
(PDF)

**S2 File. Primary reporting of studies.**
(XLSX)

## Acknowledgments

We thank Robert Wyatt from Matching Mole for help with data analysis.

## Author Contributions

**Conceptualization:** Ewoud Ewing, Benjamin Victor Ineichen.

**Data curation:** Wolfgang Emanuel Zurrer, Amelia Elaine Cannon, David Brüschweiler, Julia Bugajska, Bernard Friedrich Hild, Benjamin Victor Ineichen.

**Formal analysis:** Benjamin Victor Ineichen.

**Methodology:** Ewoud Ewing, Benjamin Victor Ineichen.

**Supervision:** Benjamin Victor Ineichen.

**Validation:** Daniel Salo Reich.

**Visualization:** Marianna Rosso.

**Writing – original draft:** Benjamin Victor Ineichen.

**Writing – review & editing:** Wolfgang Emanuel Zurrer, Amelia Elaine Cannon, Ewoud Ewing, David Brüschweiler, Julia Bugajska, Bernard Friedrich Hild, Marianna Rosso, Daniel Salo Reich.

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
