## [Decision Letter · Decision Letter 0]

5 Mar 2024

PONE-D-23-39345STEED: A data mining tool for automated extraction of experimental parameters and risk of bias items from in vivo publicationsPLOS ONE

Dear Dr. Ineichen,

Thank you for submitting your manuscript to PLOS ONE. After careful consideration, we feel that it has merit but does not fully meet PLOS ONE’s publication criteria as it currently stands. Therefore, we invite you to submit a revised version of the manuscript that addresses the points raised during the review process.

We look forward to receiving your revised manuscript.

Kind regards,

John Blake, PhD

Academic Editor

PLOS ONE

Journal Requirements:

"This work was supported by grants of the Swiss National Science Foundation (No. P400PM_183884, to BVI), the UZH Alumni (to BVI), and the Intramural Research Program of NINDS. We thank all our funders for their support."

5. Please note that funding information should not appear in the Acknowledgments section or other areas of your manuscript. We will only publish funding information present in the Funding Statement section of the online submission form. Please remove any funding-related text from the manuscript. 

**Additional Editor Comments:**

As the editor and the second reviewer, I agree with the comments made by Reviewer 1. Please ensure that you address each of the specific suggestions in your revised version.

Reviewers' comments:

Reviewer's Responses to Questions

**Comments to the Author**

1. Is the manuscript technically sound, and do the data support the conclusions?

Reviewer #1: Yes

2. Has the statistical analysis been performed appropriately and rigorously? 

Reviewer #1: Yes

3. Have the authors made all data underlying the findings in their manuscript fully available?

Reviewer #1: Yes

4. Is the manuscript presented in an intelligible fashion and written in standard English?

Reviewer #1: Yes

5. Review Comments to the Author

Reviewer #1: Thank you for the opportunity to review the manuscript titled “STEED: A data mining tool for automated extraction of experimental parameters and risk of bias items from in vivo publications.

The manuscript is concise, and clearly written. I am not a programmer but rather a potential “user” of the developed R package. Nevertheless, I found the writing to be easy to follow. The authors are open about the potential advantages as well as limitations of their software.

The supplementary materials are very helpful, especially the reporting table to give readers an idea of what the end product looks like. I have very little in terms of criticism.

L 26 In the abstract, you might want to consider using “feasibility” instead of “applicability”. Systematic reviews are still applicable, especially when there is a large body of literature, but they may not be feasible to undertake because of the labor involved.

In the discussion section, could you comment on the potential to adapt the package to other topics? Do you foresee separate packets to be developed for each discipline through adaptation of the regular expressions?

I agree with the authors that the tool may be able to at least eliminate one reviewer, while not quite replace all human reviewers yet.

Well done!

6. PLOS authors have the option to publish the peer review history of their article (what does this mean?). If published, this will include your full peer review and any attached files.

Reviewer #1: No

---

## [Author Response · Author response to Decision Letter 0]

15 Mar 2024

Point-by-point response

Copied from the decision letter, responses in blue.

Comments made by the Editor

Thank you for taking time to edit and review our manuscript.

We formatted our manuscript to meet PloS One’s style requirements. 

All custom-made code is freely and fully available at our GitHub page (https://github.com/Ineichen-Group/Auto-STEED), as stated in the manuscript (line 100).

We will submit the correct funding information.

"This work was supported by grants of the Swiss National Science Foundation (No. P400PM_183884, to BVI), the UZH Alumni (to BVI), and the Intramural Research Program of NINDS. We thank all our funders for their support."

We added this statement to the cover letter.

5. Please note that funding information should not appear in the Acknowledgments section or other areas of your manuscript. We will only publish funding information present in the Funding Statement section of the online submission form. Please remove any funding-related text from the manuscript. 

We removed this section from the manuscript.

We are now including a caption for our supporting data (i.e., the supplementary reference list) at the end of the manuscript.

We included one additional reference (see below).

8. As the editor and the second reviewer, I agree with the comments made by Reviewer 1. Please ensure that you address each of the specific suggestions in your revised version.

See our comments for R1 below. Thank you again for taking time to edit and review our manuscript.

Reviewer #1

Reviewer #1: Thank you for the opportunity to review the manuscript titled “STEED: A data mining tool for automated extraction of experimental parameters and risk of bias items from in vivo publications.

The manuscript is concise, and clearly written. I am not a programmer but rather a potential “user” of the developed R package. Nevertheless, I found the writing to be easy to follow. The authors are open about the potential advantages as well as limitations of their software.

The supplementary materials are very helpful, especially the reporting table to give readers an idea of what the end product looks like. I have very little in terms of criticism.

L 26 In the abstract, you might want to consider using “feasibility” instead of “applicability”. Systematic reviews are still applicable, especially when there is a large body of literature, but they may not be feasible to undertake because of the labor involved.

Thank you for reviewing our manuscript and for the positive and constructive comments.

We agree with this suggestion and adjusted it accordingly.

In the discussion section, could you comment on the potential to adapt the package to other topics? Do you foresee separate packets to be developed for each discipline through adaptation of the regular expressions?

This is an important point you raise. We amended the first paragraph of the discussion to discuss this in more detail which now reads:

“STEED performed well on literature corpora outside of the field is has been developed in as well as in corpora with older publication years, i.e., it has been developed in a corpus covering the motor neuron disease literature and performed well in a corpus of the multiple sclerosis literature. Thus, our devel-oped function could be applied to literature bodies of other research fields. However, adapting STEED to new disciplines requires some consideration: While the tool has shown flexibility across related fields, creating discipline-specific versions may necessitate refining the underlying RegExes to accurately capture more distinct experimental parameters pertinent to each field. This process would involve collaborative efforts with domain experts to ensure the tool's precision and subsequent valida-tion [19]. Consequently, while separate packages for each discipline are conceivable, they would re-quire some adaptation efforts to maintain STEED's standards of accuracy and utility.” (Page 14).

We also added this reference:

19. Mohammadi E, Karami A. Exploring research trends in big data across disciplines: A text mining analysis. Journal of Information Science. 2022;48(1):44-56.

I agree with the authors that the tool may be able to at least eliminate one reviewer, while not quite replace all human reviewers yet.

Well done!

Again, thank you for taking time to review our manuscript.

---

## [Decision Letter · Decision Letter 1]

18 Sep 2024

STEED: A data mining tool for automated extraction of experimental parameters and risk of bias items from in vivo publications

PONE-D-23-39345R1

Dear Dr. Ineichen,

We’re pleased to inform you that your manuscript has been judged scientifically suitable for publication and will be formally accepted for publication once it meets all outstanding technical requirements.

Kind regards,

John Blake, PhD

Academic Editor

PLOS ONE

Additional Editor Comments (optional):

I would like to clarify that I am Reviewer #2.

Reviewers' comments:

Reviewer's Responses to Questions

**Comments to the Author**

1. If the authors have adequately addressed your comments raised in a previous round of review and you feel that this manuscript is now acceptable for publication, you may indicate that here to bypass the “Comments to the Author” section, enter your conflict of interest statement in the “Confidential to Editor” section, and submit your "Accept" recommendation.

Reviewer #2: All comments have been addressed

2. Is the manuscript technically sound, and do the data support the conclusions?

Reviewer #2: Yes

3. Has the statistical analysis been performed appropriately and rigorously? 

Reviewer #2: Yes

4. Have the authors made all data underlying the findings in their manuscript fully available?

Reviewer #2: Yes

5. Is the manuscript presented in an intelligible fashion and written in standard English?

Reviewer #2: Yes

6. Review Comments to the Author

Reviewer #2: The authors have fully addressed all the comments raised in the initial round of reviews. I therefore have no further objections to publication and look forward to reading this article in print in PLOS ONE.

Comments and responses from Review 1

---

L 26 In the abstract, you might want to consider using “feasibility” instead of

“applicability”. Systematic reviews are still applicable, especially when there is a large

body of literature, but they may not be feasible to undertake because of the labor

involved.

Thank you for reviewing our manuscript and for the positive and constructive

comments. We agree with this suggestion and adjusted it accordingly.

---

In the discussion section, could you comment on the potential to adapt the package to

other topics? Do you foresee separate packets to be developed for each discipline

through adaptation of the regular expressions?

This is an important point you raise. We amended the first paragraph of the discussion

to discuss this in more detail which now reads:

“STEED performed well on literature corpora outside of the field is has been developed

in as well as in corpora with older publication years, i.e., it has been developed in a

corpus covering the motor neuron disease literature and performed well in a corpus of

the multiple sclerosis literature. Thus, our devel-oped function could be applied to

literature bodies of other research fields. However, adapting STEED to new disciplines

requires some consideration: While the tool has shown flexibility across related fields,

creating discipline-specific versions may necessitate refining the underlying RegExes

to accurately capture more distinct experimental parameters pertinent to each field.

This process would involve collaborative efforts with domain experts to ensure the

tool's precision and subsequent valida-tion [19]. Consequently, while separate

packages for each discipline are conceivable, they would re-quire some adaptation

efforts to maintain STEED's standards of accuracy and utility.” (Page 14).

----

7. PLOS authors have the option to publish the peer review history of their article (what does this mean?). If published, this will include your full peer review and any attached files.

Reviewer #2: No

---

## [Editor Report · Acceptance letter]

23 Sep 2024

PONE-D-23-39345R1 

PLOS ONE

Dear Dr. Ineichen, 

I'm pleased to inform you that your manuscript has been deemed suitable for publication in PLOS ONE. Congratulations! Your manuscript is now being handed over to our production team.

Kind regards, 

on behalf of

Dr. John Blake 

Academic Editor

PLOS ONE